# Parsimonious Bayesian deep networks

**Mingyuan Zhou**
Department of IROM, McCombs School of Business
The University of Texas at Austin, Austin, TX 78712
`mingyuan.zhou@mccombs.utexas.edu`

## Abstract

Combining Bayesian nonparametrics and a forward model selection strategy, we construct parsimonious Bayesian deep networks (PBDNs) that infer capacity-regularized network architectures from the data and require neither cross-validation nor fine-tuning when training the model. One of the two essential components of a PBDN is the development of a special infinite-wide single-hidden-layer neural network, whose number of active hidden units can be inferred from the data. The other one is the construction of a greedy layer-wise learning algorithm that uses a forward model selection criterion to determine when to stop adding another hidden layer. We develop both Gibbs sampling and stochastic gradient descent based maximum a posteriori inference for PBDNs, providing state-of-the-art classification accuracy and interpretable data subtypes near the decision boundaries, while maintaining low computational complexity for out-of-sample prediction.

## 1   Introduction

To separate two linearly separable classes, a simple linear classifier such logistic regression will often suffice, in which scenario adding the capability to model nonlinearity not only complicates the model and increases computation, but also often harms rather improves the performance by increasing the risk of overfitting. On the other hand, for two classes not well separated by a single hyperplane, a linear classifier is often inadequate, and hence it is common to use either kernel support vector machines [1, 2] or deep neural networks [3–5] to nonlinearly transform the covariates, making the two classes become more linearly separable in the transformed covariate space. While being able to achieve high classification accuracy, they both have clear limitations. For a kernel based classifier, its number of support vectors often increases linearly in the size of training data [6], making it not only computationally expensive and memory inefficient to train for big data, but also slow in out-of-sample predictions. A deep neural network could be scalable with an appropriate network structure, but it is often cumbersome to tune the network depth (number of layers) and width (number of hidden units) of each hidden layer [5], and has the danger of overfitting the training data if a deep neural network, which is often equipped with a larger than necessary modeling capacity, is not carefully regularized.

Rather than making an uneasy choice in the first place between a linear classifier, which has fast computation and resists overfitting but may not provide sufficient class separation, and an over-capacitized model, which often wastes computation and requires careful regularization to prevent overfitting, we propose a parsimonious Bayesian deep network (PBDN) that builds its capacity regularization into the greedy-layer-wise construction and training of the deep network. More specifically, we transform the covariates in a layer-wise manner, with each layer of transformation designed to facilitate class separation via the use of the noisy-OR interactions of multiple weighted linear hyperplanes. Related to kernel support vector machines, the hyperplanes play a similar role as support vectors in transforming the covariate space, but they are inferred from the data and their number increases at a much slower rate with the training data size. Related to deep neural networks, the proposed multi-layer structure gradually increases its modeling capability by increasing its number

of layers, but allows inferring from the data both the width of each hidden layer and depth of the network to prevent building a model whose capacity is larger than necessary.

To obtain a parsimonious deep neural network, one may also consider using a two-step approach that first trains an over-capacitized model and then compresses its size [7–10]. The design of PBDN represents a distinct philosophy. Moreover, PBDN is not contradicting with model compression, as these post-processing compression techniques [7–10] may be used to further compress PBDN.

For capacity regularization of the proposed PBDN, we choose to shrink both its width and depth. To shrink the width of a hidden layer, we propose the use of a gamma process [11], a draw from which consists of countably infinite atoms, each of which is used to represent a hyperplane in the covariate space. The gamma process has an inherence shrinkage mechanism as its number of atoms, whose random weights are larger than a certain positive constant $\epsilon > 0$, follows a Poisson distribution, whose mean is finite almost surely (a.s.) and reduces towards zero as $\epsilon$ increases. To shrink the depth of the network, we propose a layer-wise greedy-learning strategy that increases the depth by adding one hidden layer at a time, and uses an appropriate model selection criterion to decide when to stop adding another one. Note related to our work, Zhou et al. [12, 13] combine the gamma process and greedy layer-wise training to build a Bayesian deep network in an unsupervised manner. Our experiments show the proposed capacity regularization strategy helps successfully build a PBDN, providing state-of-the-art classification accuracy while maintaining low computational complexity for out-of-sample prediction. We have also tried applying a highly optimized off-the-shelf deep neural network based classifier, whose network architecture for a given data is set to be the same as that inferred by the PBDN. However, we have found no performance gains, suggesting the efficacy of the PBDN's greedy training procedure that requires neither cross-validation nor fine-tuning. Note PBDN, like a conventional deep neural network, could also be further improved by introducing convolutional layers if the covariates have spatial, temporal, or other types of structures.

## 2    Layer-width learning via infinite support hyperplane machines

The first essential component of the proposed capacity regularization strategy is to learn the width of a hidden layer. To fulfill this goal, we define infinite support hyperplane machine (iSHM), a label-asymmetric classifier, that places in the covariate space countably infinite hyperplanes $\{\boldsymbol{\beta}_k\}_{1,\infty}$, where each $\boldsymbol{\beta}_k \in \mathbb{R}^{V+1}$ is associated with a weight $r_k > 0$. We use a gamma process [11] to generate $\{r_k, \boldsymbol{\beta}_k\}_{1,\infty}$, making the infinite sum $\sum_{k=1}^{\infty} r_k$ be finite almost surely (a.s.). We measure the proximity of a covariate vector $\boldsymbol{x}_i \in \mathbb{R}^{V+1}$ to $\boldsymbol{\beta}_k$ using the softplus function of their inner product as $\ln(1 + e^{\boldsymbol{\beta}_k' \boldsymbol{x}_i})$, which is a smoothed version of $\text{ReLU}(\boldsymbol{\beta}_k' \boldsymbol{x}_i) = \max(0, \boldsymbol{\beta}_k' \boldsymbol{x}_i)$ that is widely used in deep neural networks [14–17]. We consider that $\boldsymbol{x}_i$ is far from hyperplane $k$ if $\ln(1+e^{\boldsymbol{\beta}_k' \boldsymbol{x}_i})$ is close to zero. Thus, as $\boldsymbol{x}_i$ moves away from hyperplane $k$, that proximity measure monotonically increases on one side of the hyperplane while decreasing on the other. We pass $\lambda_i = \sum_{k=1}^{\infty} r_k \ln(1 + e^{\boldsymbol{\beta}_k' \boldsymbol{x}_i})$, a non-negative weighted combination of these proximity measures, through the Bernoulli-Poisson link [18] $f(\lambda_i) = 1 - e^{-\lambda_i}$ to define the conditional class probability as

$$P(y_i = 1 \,|\, \{r_k, \boldsymbol{\beta}_k\}_k, \boldsymbol{x}_i) = 1 - \prod_{k=1}^{\infty} (1 - p_{ik}), \qquad p_{ik} = 1 - e^{-r_k \ln(1+e^{\boldsymbol{\beta}_k' \boldsymbol{x}_i})}. \tag{1}$$

Note the model treats the data labeled as "0" and "1" differently, and (1) suggests that in general $P(y_i = 1 \,|\, \{r_k, \boldsymbol{\beta}_k\}_k, \boldsymbol{x}_i) \neq 1 - P(y_i = 0 \,|\, \{r_k, -\boldsymbol{\beta}_k\}_k, \boldsymbol{x}_i)$. We will show that

$$\sum_i p_{ik} \boldsymbol{x}_i / \sum_i p_{ik} \tag{2}$$

can be used to represent the $k$th data subtype discovered by the algorithm.

### 2.1    Nonparametric Bayesian hierarchical model

One may readily notice from (1) that the noisy-OR construction, widely used in probabilistic reasoning [19–21], is generalized by iSHM to attribute a binary outcome of $y_i = 1$ to countably infinite hidden causes $p_{ik}$. Denoting $\bigvee$ as the logical OR operator, as $P(y = 0) = \prod_k P(b_k = 0)$ if $y = \bigvee_k b_k$ and $\mathbb{E}[e^{-\theta}] = e^{-r \ln(1+e^x)}$ if $\theta \sim \text{Gamma}(r, e^x)$, we have an augmented form of (1) as

$$y_i = \bigvee_{k=1}^{\infty} b_{ik}, \;\; b_{ik} \sim \text{Bernoulli}(p_{ik}), \; p_{ik} = 1 - e^{-\theta_{ik}}, \; \theta_{ik} \sim \text{Gamma}(r_k, e^{\boldsymbol{\beta}_k' \boldsymbol{x}_i}), \tag{3}$$

where $b_{ik} \sim \text{Bernoulli}(p_{ik})$ can be further augmented as $b_{ik} = \delta(m_{ik} \geq 1)$, $m_{ik} \sim \text{Pois}(\theta_{ik})$, where $m_{ik} \in \mathbb{Z}$, $\mathbb{Z} := \{0, 1, \ldots\}$, and $\delta(x)$ equals to 1 if the condition $x$ is satisfied and 0 otherwise.

We now marginalize $b_{ik}$ out to formally define iSHM. Let $G \sim \Gamma\mathrm{P}(G_0, 1/c)$ denote a gamma process defined on the product space $\mathbb{R}_+ \times \Omega$, where $\mathbb{R}_+ = \{x : x > 0\}$, $c \in \mathbb{R}_+$, and $G_0$ is a finite and continuous base measure over a complete separable metric space $\Omega$. As illustrated in Fig. 2 (b) in the Appendix, given a draw from $G$, expressed as $G = \sum_{k=1}^{\infty} r_k \delta_{\boldsymbol{\beta}_k}$, where $\boldsymbol{\beta}_k$ is an atom and $r_k$ is its weight, the iSHM generates the label under the Bernoulli-Poisson link [18] as

$$y_i \,|\, G, \boldsymbol{x}_i \sim \mathrm{Bernoulli}\big(1 - e^{-\sum_{k=1}^{\infty} r_k \ln(1 + e^{\boldsymbol{x}_i' \boldsymbol{\beta}_k})}\big), \tag{4}$$

which can be represented as a noisy-OR model as in (3) or, as shown in Fig. 2 (a), constructed as

$$y_i = \delta(m_i \geq 1), \;\; m_i = \textstyle\sum_{k=1}^{\infty} m_{ik}, \;\; m_{ik} \sim \mathrm{Pois}(\theta_{ik}), \;\; \theta_{ik} \sim \mathrm{Gamma}(r_k, e^{\boldsymbol{\beta}_k' \boldsymbol{x}_i}). \tag{5}$$

From (3) and (5), it is clear that one may declare hyperplane $k$ as inactive if $\sum_i b_{ik} = \sum_i m_{ik} = 0$.

## 2.2 Inductive bias and distinction from multilayer perceptron

Below we reveal the inductive bias of iSHM in prioritizing the fit of the data labeled as "1," due to the use of the Bernoulli-Poisson link that has previously been applied for network analysis [18, 22, 23] and multi-label learning [24]. As the negative log-likelihood (NLL) for $\boldsymbol{x}_i$ can be expressed as

$$\mathrm{NLL}(\boldsymbol{x}_i) = -y_i \ln\big(1 - e^{-\lambda_i}\big) + (1 - y_i)\lambda_i, \;\; \lambda_i = \textstyle\sum_{k=1}^{\infty} r_k \ln(1 + e^{\boldsymbol{x}_i' \boldsymbol{\beta}_k}),$$

we have $\mathrm{NLL}(\boldsymbol{x}_i) = \lambda_i - \ln(e^{\lambda_i} - 1)$ if $y_i = 1$ and $\mathrm{NLL}(\boldsymbol{x}_i) = \lambda_i$ if $y_i = 0$. As $-\ln(e^{\lambda_i} - 1)$ quickly explodes towards $+\infty$ as $\lambda_i \to 0$, when $y_i = 1$, iSHM would adjust $r_k$ and $\boldsymbol{\beta}_k$ to avoid at all cost overly suppressing $\boldsymbol{x}_i$ (i.e., making $\lambda_i$ too small). By contrast, it has a high tolerance of failing to sufficiently suppress $\boldsymbol{x}_i$ with $y_i = 0$. Thus each $\boldsymbol{x}_i$ with $y_i = 1$ would be made sufficiently close to at least one active support hyperplane. By contrast, while each $\boldsymbol{x}_i$ with $y_i = 0$ is desired to be far away from any support hyperplanes, violating that is typically not strongly penalized. Therefore, by training a pair of iSHMs under two opposite labeling settings, two sets of support hyperplanes could be inferred to sufficiently cover the covariate space occupied by the training data from both classes.

Note as in (4), iSHM may be viewed as an infinite-wide single-hidden-layer neural network that connects the input layer to the $k$th hidden unit via the connections weights $\boldsymbol{\beta}_k$ and the softplus nonlinear activation function $\ln(1 + e^{\boldsymbol{\beta}_k' \boldsymbol{x}_i})$, and further pass a non-negative weighted combination of these hidden units through the Bernoulli-Poisson link to obtain the conditional class probability. From this point of view, it can be related to a single-hidden-layer multilayer perceptron (MLP) [5, 25] that uses a softplus activation function and cross-entropy loss, with the output activation expressed as $\sigma[\boldsymbol{w}' \ln(1 + e^{\mathbf{B}\boldsymbol{x}_i})]$, where $\sigma(x) = 1/(1 + e^{-x})$, $K$ is the number of hidden units, $\mathbf{B} = (\boldsymbol{\beta}_1, \ldots, \boldsymbol{\beta}_K)'$, and $\boldsymbol{w} = (w_1, \ldots, w_K)' \in \mathbb{R}^K$. Note minimizing the cross-entropy loss is equivalent to maximizing the likelihood of $y_i \,|\, \boldsymbol{w}, \mathbf{B}, \boldsymbol{x}_i \sim \mathrm{Bernoulli}\big[(1 + e^{-\sum_{k=1}^{K} w_k \ln(1 + e^{\boldsymbol{x}_i' \boldsymbol{\beta}_k})})^{-1}\big]$, which is biased towards fitting neither the data with $y_i = 1$ nor these with $y_i = 0$, since

$$\mathrm{NLL}(\boldsymbol{x}_i) = \ln(e^{-y_i \boldsymbol{w}' \ln(1 + e^{\mathbf{B}\boldsymbol{x}_i})} + e^{(1 - y_i)\boldsymbol{w}' \ln(1 + e^{\mathbf{B}\boldsymbol{x}_i})}).$$

Therefore, while iSHM is structurally similar to an MLP, it is distinct in its unbounded layer width, its positive constraint on the weights $r_k$ connecting the hidden and output layers, its ability to rigorously define whether a hyperplane is active or inactive, and its inductive bias towards fitting the data labeled as "1." As in practice labeling which class as "1" may be arbitrary, we predict the class label with $(1 - e^{-\sum_{k=1}^{\infty} r_k \ln(1 + e^{\boldsymbol{x}_i' \boldsymbol{\beta}_k})} + e^{-\sum_{k=1}^{\infty} r_k^* \ln(1 + e^{\boldsymbol{x}_i' \boldsymbol{\beta}_k^*})})/2$, where $\{r_k, \boldsymbol{\beta}_k\}_{1,\infty}$ and $\{r_k^*, \boldsymbol{\beta}_k^*\}_{1,\infty}$ are from a pair of iSHMs trained by labeling the data belonging to this class as "1" and "0," respectively.

## 2.3 Convex polytope geometric constraint

It is straightforward to show that iSHM with a single unit-weighted hyperplane reduces to logistic regression $y_i \sim \mathrm{Bernoulli}[1/(1 + e^{-\boldsymbol{x}_i' \boldsymbol{\beta}})]$. To interpret the role of each individual support hyperplane when multiple non-negligibly weighted ones are inferred by iSHM, we analogize each $\boldsymbol{\beta}_k$ to an expert of a committee that collectively make binary decisions. For expert (hyperplane) $k$, the weight $r_k$ indicates how strongly its opinion is weighted by the committee, $b_{ik} = 0$ represents that it votes "No," and $b_{ik} = 1$ represents that it votes "Yes." Since $y_i = \bigvee_{k=1}^{\infty} b_{ik}$, the committee would vote "No" if and only if all its experts vote "No" (i.e., all $b_{ik}$ are zeros), in other words, the committee would vote "Yes" even if only a single expert votes "Yes." Let us now examine the confined covariate space that

satisfies the inequality $P(y_i = 1 \,|\, \boldsymbol{x}_i) \leq p_0$, where a data point is labeled as "1" with a probability no greater than $p_0$. The following theorem shows that it defines a confined space bounded by a convex polytope, as defined by the intersection of countably infinite half-spaces defined by $p_{ik} < p_0$.

**Theorem 1** (Convex polytope). *For iSHM, the confined space specified by the inequality*

$$P(y_i = 1 \,|\, \{r_k, \boldsymbol{\beta}_k\}_k, \boldsymbol{x}_i) \leq p_0 \tag{6}$$

*is bounded by a convex polytope defined by the set of solutions to countably infinite inequalities as*

$$\boldsymbol{x}_i'\boldsymbol{\beta}_k \leq \ln\left[(1 - p_0)^{-\frac{1}{r_k}} - 1\right], \; k \in \{1, 2, \ldots\}. \tag{7}$$

The convex polytope defined in (7) is enclosed by the intersection of countably infinite half-spaces. If we set $p_0 = 0.5$ as the probability threshold to make binary decisions, then the convex polytope assigns a label of $y_i = 0$ to an $\boldsymbol{x}_i$ inside the convex polytope (*i.e.*, an $\boldsymbol{x}_i$ that satisfies all the inequalities in Eq. 7) with a relatively high probability, and assigns a label of $y_i = 1$ to an $\boldsymbol{x}_i$ outside the convex polytope (*i.e.*, an $\boldsymbol{x}_i$ that violates at least one of the inequalities in Eq. 7) with a probability of at least 50%. Note that hyperplane $k$ with $r_k \to 0$ has a negligible impact on the conditional class probability. Choosing the gamma process as the nonparametric Bayesian prior sidesteps the need to tune the number of experts. It shrinks the weights of all unnecessary experts, allowing automatically inferring a finite number of non-negligibly weighted ones (support hyperplanes) from the data. We provide in Appendix B the connections to previously proposed multi-hyperplane models [26–30].

## 2.4 Gibbs sampling and MAP inference via SGD

For the convenience of implementation, we truncate the gamma process with a finite and discrete base measure as $G_0 = \sum_{k=1}^{K} \frac{\gamma_0}{K} \delta_{\boldsymbol{\beta}_k}$, where $K$ will be set sufficiently large to approximate the truly countably infinite model. We express iSHM using (5) together with

$$r_k \sim \text{Gamma}(\gamma_0/K, 1/c_0), \; \gamma_0 \sim \text{Gamma}(a_0, 1/b_0), \; c_0 \sim \text{Gamma}(e_0, 1/f_0),$$

$$\boldsymbol{\beta}_k \sim \prod_{v=0}^{V} \int \mathcal{N}(0, \alpha_{vk}^{-1})\text{Gamma}(\alpha_{vk}; a_\beta, 1/b_{\beta k}) d\alpha_{vk}, \; b_{\beta k} \sim \text{Gamma}(e_0, 1/f_0),$$

where the normal gamma construction promotes sparsity on the connection weights $\boldsymbol{\beta}_k$ [31].

We describe both Gibbs sampling, desirable for uncertainty quantification, and maximum a posteriori (MAP) inference, suitable for large-scale training, in Algorithm 1. We use data augmentation and marginalization to derive Gibbs sampling, with the details deferred to Appendix B. For MAP inference, we use Adam [32] in Tensorflow to minimize a stochastic objective function as $f(\{\boldsymbol{\beta}_k, \ln r_k\}_1^K, \{y_i, \boldsymbol{x}_i\}_{i_1}^{i_M}) + f(\{\boldsymbol{\beta}_k^*, \ln r_k^*\}_1^{K^*}, \{y_i^*, \boldsymbol{x}_i\}_{i_1}^{i_M})$, which embeds the hierarchical Bayesian model's inductive bias and inherent shrinking mechanism into optimization, where $M$ is the size of a randomly selected mini-batch, $y_i^* := 1 - y_i$, $\lambda_i := \sum_{k=1}^{K} e^{\ln r_k} \ln(1 + e^{\boldsymbol{x}_i'\boldsymbol{\beta}_k})$, and

$$f(\{\boldsymbol{\beta}_k, \ln r_k\}_1^K, \{y_i, \boldsymbol{x}_i\}_{i_1}^{i_M}) = \sum_{k=1}^{K}\left(-\tfrac{\gamma_0}{K}\ln r_k + c_0 e^{\ln r_k}\right) + (a_\beta + 1/2)\sum_{v=0}^{V}\sum_{k=0}^{K}$$
$$[\ln(1 + \beta_{vk}^2/(2b_{\beta k}))] + \tfrac{N}{M}\sum_{i=i_1}^{i_M}\left[-y_i \ln\left(1 - e^{-\lambda_i}\right) + (1 - y_i)\lambda_i\right]. \tag{8}$$

## 3 Network-depth learning via forward model selection

The second essential component of the proposed capacity regularization strategy is to find a way to increase the network depth and determine how deep is deep enough. Our solution is to sequentially stack a pair of iSHMs on top of the previously trained one, and develop a forward model selection criterion to decide when to stop stacking another pair. We refer to the resulted model as parsimonious Bayesian deep network (PBDN), as described below in detail.

The noisy-OR hyperplane interactions allow iSHM to go beyond simple linear separation, but with limited capacity due to the convex-polytope constraint imposed on the decision boundary. On the other hand, it is the convex-polytope constraint that provides an implicit regularization, determining how many non-negligibly weighted support hyperplanes are necessary in the covariate space to sufficiently activate all data of class "1," while somewhat suppressing the data of class "0." In this paper, we find that the model capacity could be quickly enhanced by sequentially stacking such convex-polytope constraints under a feedforward deep structure, while preserving the virtue of being able to learn the number of support hyperplanes in the (transformed) covariate space.

More specifically, as shown in Fig. 2 (c) of the Appendix, we first train a pair of iSHMs that regress the current labels $y_i \in \{0, 1\}$ and the flipped ones $y_i^* = 1 - y_i$, respectively, on the original covariates $\boldsymbol{x}_i \in \mathbb{R}^V$. After obtaining $K_2$ support hyperplanes $\{\boldsymbol{\beta}_k^{(1 \to 2)}\}_{1, K_2}$, constituted by the active support hyperplanes inferred by both iSHM trained with $y_i$ and the one trained with $y_i^*$, we use $\ln(1 + e^{\boldsymbol{x}_i' \boldsymbol{\beta}_k^{(1 \to 2)}})$ as the hidden units of the second layer (first hidden layer). More precisely, with $t \in \{1, 2, \ldots\}$, $K_0 := 0$, $K_1 := V$, $\tilde{\boldsymbol{x}}_i^{(0)} := \emptyset$, and $\tilde{\boldsymbol{x}}_i^{(1)} := \boldsymbol{x}_i$, denoting $\boldsymbol{x}_i^{(t)} := [1, (\tilde{\boldsymbol{x}}_i^{(t-1)})', (\tilde{\boldsymbol{x}}_i^{(t)})']' \in \mathbb{R}^{K_{t-1} + K_t + 1}$ as the input data vector to layer $t + 1$, the $t$th added pair of iSHMs transform $\boldsymbol{x}_i^{(t)}$ into the hidden units of layer $t + 1$, expressed as

$$\tilde{\boldsymbol{x}}_i^{(t+1)} = \left[ \ln(1 + e^{(\boldsymbol{x}_i^{(t)})' \boldsymbol{\beta}_1^{(t \to t+1)}}), \ldots, \ln(1 + e^{(\boldsymbol{x}_i^{(t)})' \boldsymbol{\beta}_{K_{t+1}}^{(t \to t+1)}}) \right]' .$$

Hence the input vectors used to train the next layer would be $\boldsymbol{x}_i^{(t+1)} = [1, (\tilde{\boldsymbol{x}}_i^{(t)})', (\tilde{\boldsymbol{x}}_i^{(t+1)})']' \in \mathbb{R}^{K_t + K_{t+1} + 1}$. Therefore, if the computational cost of a single inner product $\boldsymbol{x}_i' \boldsymbol{\beta}_k$ (e.g., logistic regression) is one, then that for $T$ hidden layers would be about $\sum_{t=1}^T (K_{t-1} + K_t + 1) K_{t+1} / (V + 1)$. Note one may also use $\boldsymbol{x}_i^{(t+1)} = \tilde{\boldsymbol{x}}_i^{(t+1)}$, or $\boldsymbol{x}_i^{(t+1)} = [(\boldsymbol{x}_i^{(t)})', (\tilde{\boldsymbol{x}}_i^{(t+1)})']'$, or other related concatenation methods to construct the covariates to train the next layer.

Our intuition for why PBDN, constructed in this greedy-layer-wise manner, works well is that for two iSHMs trained on the same covariate space under two opposite labeling settings, one iSHM places enough hyperplanes to define the complement of a convex polytope to sufficiently activate all data labeled as "1," while the other does so for all data labeled as "0." Thus, for any $\boldsymbol{x}_i$, at least one $p_{ik}$ would be sufficiently activated, in other words, $\boldsymbol{x}_i$ would be sufficiently close to at least one of the active hyperplanes of the iSHM pair. This mechanism prevents any $\boldsymbol{x}_i$ from being completely suppressed after transformation. Consequently, these transformed covariates $\ln(1 + e^{\boldsymbol{\beta}_k' \boldsymbol{x}_i})$, which can also be concatenated with $\boldsymbol{x}_i$, will be further used to train another iSHM pair. Thus even though a single iSHM pair may not be powerful enough, by keeping all covariate vectors sufficiently activated after transformation, they could be simply stacked sequentially to gradually enhance the model capacity, with a strong resistance to overfitting and hence without the necessity of cross-validation.

While stacking an additional iSHM pair on PBDN could enhance the model capacity, when $T$ hidden layers is sufficiently deeper that the two classes become well separated, there is no more need to add an extra iSHM pair. To detect when it is appropriate to stop adding another iSHM pair, as shown in Algorithm 2 of the Appendix, we consider a forward model selection strategy that sequentially stacks an iSHM pair after another, until the following criterion starts to rise:

$$\text{AIC}(T) = \sum_{t=1}^T [2(K_t + 1)K_{t+1}] + 2K_{T+1} - 2 \sum_i \left[ \ln P(y_i \mid \boldsymbol{x}_i^{(T)}) + \ln P(y_i^* \mid \boldsymbol{x}_i^{(T)}) \right], \quad (9)$$

where $2(K_t + 1)K_{t+1}$ represents the cost of adding the $t$th hidden layer and $2K_{T+1}$ represents the cost of using $K_{T+1}$ nonnegative weights $\{r_k^{(T+1)}\}_k$ to connect the $T$th hidden layer and the output layer. With (9), we choose PBDN with $T$ hidden layers if $\text{AIC}(t+1) \le \text{AIC}(t)$ for $t = 1, \ldots, T-1$ and $\text{AIC}(T+1) > \text{AIC}(T)$. We also consider another model selection criterion that accounts for the sparsity of $\boldsymbol{\beta}_k^{(t \to t+1)}$, the connection weights between adjacent layers, using

$$\text{AIC}_\epsilon(T) = \sum_{t=1}^T 2 \left( \||\mathbf{B}_t| > \epsilon \beta_{t\max}\|_0 + \||\mathbf{B}_t^*| > \epsilon \beta_{t\max}^*\|_0 \right) + 2K_{T+1} - 2 \sum_i \left[ \ln P(y_i \mid \boldsymbol{x}_i^{(T)}) + \ln P(y_i^* \mid \boldsymbol{x}_i^{(T)}) \right], \quad (10)$$

where $\|\mathbf{B}\|_0$ is the number of nonzero elements in matrix $\mathbf{B}$, $\epsilon > 0$ is a small constant, and $\mathbf{B}_t$ and $\mathbf{B}_t^*$ consist of the $\boldsymbol{\beta}_k^{(t \to t+1)}$ trained by the first and second iSHMs of the $t$th iSHM pair, respectively, with $\beta_{t\max}$ and $\beta_{t\max}^*$ as their respective maximum absolute values.

Note if the covariates have any spatial or temporal structures to exploit, one may replace each $\boldsymbol{x}_i' \boldsymbol{\beta}_k$ in iSHM with $[\text{CNN}_\phi(\boldsymbol{x}_i)]' \boldsymbol{\beta}_k$, where $\text{CNN}_\phi(\cdot)$ represents a convolutional neural network parameterized by $\phi$, to construct a CNN-iSHM, which can then be further greedily grown into CNN-PBDN. Customizing PBDNs for structured covariates, such as image pixels and audio measurements, is a promising research topic, however, is beyond the scope of this paper.

## 4 Illustrations and experimental results

Code for reproducible research is available at https://github.com/mingyuanzhou/PBDN. To illustrate the imposed geometric constraint and inductive bias of a single iSHM, we first consider a challenging

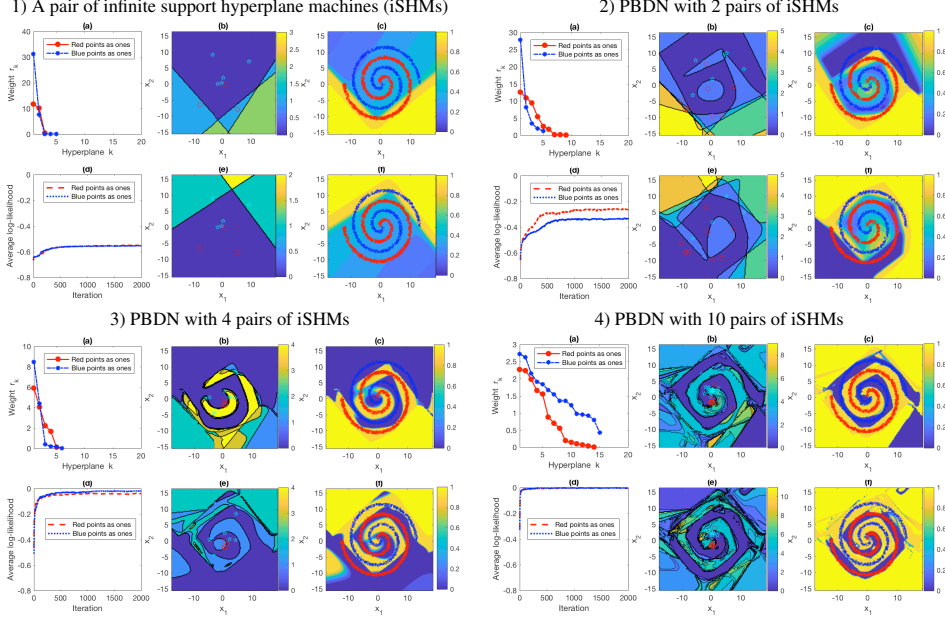

Figure 1: Visualization of PBDN, each layer of which is a pair of iSHMs trained on a "two spirals" dataset under two opposite labeling settings. For **Subfigure 1)**, (a) shows the weights $r_k$ of the inferred active support hyperplanes, ordered by their values and (d) shows the trace plot of the average per data point log-likelihood. For iSHM trained by labeling the red (blue) points as ones (zeros), (b) shows a contour map, the value of each point of which represents how many inequalities specified in (7) are violated, and whose region with zero values corresponds to the convex polytope enclosed by the intersection of the hyperplanes defined in (7), and (c) shows the contour map of predicted class probabilities. (e-f) are analogous plots to (b-c) for iSHM trained with the blue points labeled as ones. The inferred data subtypes as in (2) are represented as red circles and blue pentagrams in subplots (b) and (d). **Subfigures 2)-4)** are analogous to 1), with two main differences: ($i$) the transformed covariates to train the newly added iSHM pair are obtained by propagating the original 2-D covariates through the previously trained iSHM pairs, and ($ii$) the contour maps in subplots (b) and (e) visualize the iSHM linear hyperplanes in the transformed space by projecting them back to the original 2-D covariate space.

2-D "two spirals" dataset, as shown Fig. 1, whose two classes are not fully separable by a convex polytope. We train 10 pairs of iSHMs one pair after another, which are organized into a ten-hidden-layer PBDN, whose numbers of hidden units from the 1st to 10th hidden layers (*i.e.*, numbers of support hyperplanes of the 1st to 10th iSHM pairs) are inferred to be 8, 14, 15, 11, 19, 22, 23, 18, 19, and 29, respectively. Both AIC and AIC$_{\epsilon=0.01}$ infers the depth as $T = 4$.

For Fig. 1, we first train an iSHM by labeling the red points as "1" and blue as "0," whose results are shown in subplots 1) (b-c), and another iSHM under the opposite labeling setting, whose results are shown in subplots 1) (e-f). It is evident that when labeling the red points as "1," as shown in 1) (a-c), iSHM infers a convex polytope, intersected by three active support hyperplanes, to enclose the blue points, but at the same time allows the red points within that convex polytope to pass through with appreciable activations. When labeling the blue points as "1," iSHM infers five active support hyperplane, two of which are visible in the covariate space shown in 1) (e), to enclose the red points, but at the same time allows the blue points within that convex polytope to pass through with appreciable activations, as shown in 1) (f). Only capable of using a convex polytope to enclose the data labeled as "0" is a restriction of iSHM, but it is also why the iSHM pair could appropriately place two parsimonious sets of active support hyperplanes in the covariate space, ensuring the maximum distance of any data point to these support hyperplanes to be sufficiently small.

Second, we concatenate the 8-D transformed and 2-D original covariates as 10-D covariates, which is further augmented with a constant bias term of one, to train the second pair of iSHMs. As in subplot 2) (a), five active support hyperplanes are inferred in the 10-D covariate space when labeling the blue points as "1," which could be matched to five nonzero smooth segments of the original 2-D spaces shown in 2) (e), which well align with highly activated regions in 2) (f). Again, with the inductive bias, as in 2) (f), all positive labeled data points are sufficiently activated at the expense of allowing some negative ones to be only moderately suppressed. Nine support hyperplanes are inferred when labeling the red points as "1," and similar activation behaviors can also be observed in 2) (b-c).

Table 1: Visualization of the subtypes inferred by PBDN in a random trial and comparison of classification error rates over five random trials between PBDN and a two-hidden-layer DNN (128-64) on four different MNIST binary classification tasks.

| | (a) Subtypes of 3 in 3 vs 5 | (b) Subtypes of 3 in 3 vs 8 | (c) Subtypes of 4 in 4 vs 7 | (d) Subtypes of 4 in 4 vs 9 |
|---|---|---|---|---|
| | 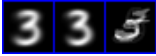 |  |  |  |
| | (e) Subtypes of 5 in 3 vs 5 | (f) Subtypes of 8 in 3 vs 8 | (g) Subtypes of 7 in 4 vs 7 | (h) Subtypes of 9 in 4 vs 9 |
| |  |  |  |  |
| PBDN | $2.53\% \pm 0.22\%$ | $2.66\% \pm 0.27\%$ | $1.37\% \pm 0.18\%$ | $2.95\% \pm 0.47\%$ |
| DNN | $2.78\% \pm 0.36\%$ | $2.93\% \pm 0.40\%$ | $1.21\% \pm 0.12\%$ | $2.98\% \pm 0.17\%$ |

We continue the same greedy layer-wise training strategy to add another eight iSHM pairs. From Figs. 1 3)-4) it becomes more and more clear that a support hyperplane, inferred in the transformed covariate space of a deep hidden layer, could be used to represent the boundary of a complex but smooth segment of the original covariate space that well encloses all or a subset of data labeled as "1."

We apply PBDN to four different MNIST binary classification tasks and compare its performance with DNN (128-64), a two-hidden-layer deep neural network that will be detailedly described below. As in Tab. 1, both AIC and AIC$_{\epsilon=0.01}$ infer the depth as $T = 1$ for PBDN, and infer for each class only a few active hyperplanes, each of which represents a distinct data subtype, as calculated with (2). In a random trial, the inferred networks of PBDN for all four tasks have only a single hidden layer with at most 6 active hidden units. Thus its testing computation is much lower than DNN (128-64), while providing an overall lower testing error rate (both trained with 4000 mini-batches of size 100).

Below we provide comprehensive comparison on eight widely used benchmark datasets between the proposed PBDNs and a variety of algorithms, including logistic regression, Gaussian radial basis function (RBF) kernel support vector machine (SVM), relevance vector machine (RVM) [31], adaptive multi-hyperplane machine (AMM) [27], convex polytope machine (CPM) [30], and the deep neural network (DNN) classifier (DNNClassifier) provided in Tensorflow [33]. Except for logistic regression that is a linear classifier, both kernel SVM and RVM are widely used nonlinear classifiers relying on the kernel trick, both AMM and CPM intersect multiple hyperplanes to construct their decision boundaries, and DNN uses a multilayer feedforward network, whose network structure often needs to be tuned to achieve a good balance between data fitting and model complexity, to handle nonlinearity. We consider DNN (8-4), a two-hidden-layer DNN that uses 8 and 4 hidden units for its first and second hidden layers, respectively, DNN (32-16), and DNN (128-64). In the Appendix, we summarize in Tab. 4 the information of eight benchmark datasets, including banana, breast cancer, titanic, waveform, german, image, ijcnn1, and a9a. For a fair comparison, to ensure the same training/testing partitions for all algorithms across all datasets, we report the results by using either widely used open-source software packages or the code made public available by the original authors. We describe in the Appendix the settings of all competing algorithms.

For all datasets, we follow Algorithm 1 to first train a single-hidden-layer PBDN (PBDN1), *i.e.*, a pair of iHSMs fitted under two opposite labeling settings. We then follow Algorithm 2 to train another pair of iHSMs to construct a two-hidden-layer PBDN (PBDN2), and repeat the same procedure to train PBDN3 and PBDN4. Note we observe that PBDN's log-likelihood increases rapidly during the first few hundred MCMC/SGD iterations, and then keeps increasing at a slower pace and eventually fluctuates. However, it often takes more iterations to shrink the weights of unneeded hyperplanes towards deactivation. Thus although insufficient iterations may not necessarily degrade the final out-of-sample prediction accuracy, it may lead to a less compact network and hence higher computational cost for out-of-sample prediction. For each iHSM, we set the upper bound on the number of support hyperplanes as $K_{\max} = 20$. For Gibbs sampling, we run 5000 iterations and record $\{r_k, \boldsymbol{\beta}_k\}_k$ with the highest likelihood during the last 2500 iterations; for MAP, we process 4000 mini-batches of size $M = 100$, with $0.05/(4 + T)$ as the Adam learning rate for the $T$th added iHSM pair. We use the inferred $\{r_k, \boldsymbol{\beta}_k\}_k$ to either produce out-of-sample predictions or generate transformed covariates for the next layer. We set $a_0 = b_0 = 0.01$, $e_0 = f_0 = 1$, and $a_\beta = 10^{-6}$ for Gibbs sampling. We fix $\gamma_0 = c_0 = 1$ and $a_\beta = b_{\beta k} = 10^{-6}$ for MAP inference. As in Algorithm 1, we prune inactive support hyperplanes once every 200 MCMC or 500 SGD iterations to facilitate computation.

Table 2: Comparison of classification error rates between a variety of algorithms and the proposed PBDNs with 1, 2, or 4 hidden layers, and PBDN-AIC and PBDN-AIC$_{\epsilon=0.01}$ trained with Gibbs sampling or SGD. Displayed in the last two rows of each column are the average of the error rates and that of computational complexities of an algorithm normalized by those of kernel SVM.

| | LR | SVM | RVM | AMM | CPM | DNN (8-4) | DNN (32-16) | DNN (128-64) | PBDN1 | PBDN2 | PBDN4 | AIC Gibbs | AIC$_\epsilon$ Gibbs | AIC SGD | AIC$_\epsilon$ SGD |
|---|---|---|---|---|---|---|---|---|---|---|---|---|---|---|---|
| banana | 47.76 ±4.38 | 10.85 ±0.57 | 11.08 ±0.69 | 18.76 ±4.09 | 21.59 ±3.00 | 14.07 ±5.87 | 10.88 ±0.36 | 10.91 ±0.45 | 22.54 ±5.28 | 10.94 ±0.46 | 10.85 ±0.43 | 10.97 ±0.46 | 10.97 ±0.46 | 11.49 ±0.65 | 11.79 ±0.89 |
| breast cancer | 28.05 ±3.68 | 28.44 ±4.52 | 31.56 ±4.66 | 31.82 ±4.47 | 30.13 ±5.26 | 32.73 ±4.77 | 33.51 ±6.47 | 32.21 ±6.64 | 29.22 ±3.53 | 30.26 ±4.86 | 30.26 ±5.55 | 29.22 ±3.53 | 29.22 ±3.53 | 32.08 ±6.18 | 29.87 ±5.27 |
| titanic | 22.67 ±0.98 | 22.33 ±0.63 | 23.20 ±1.08 | 28.85 ±8.56 | 23.38 ±3.23 | 22.32 ±1.38 | 22.35 ±1.36 | 22.28 ±1.37 | 22.88 ±0.59 | 22.25 ±1.27 | 22.16 ±1.13 | 22.88 ±0.59 | 22.88 ±0.59 | 23.00 ±0.37 | 23.00 ±0.37 |
| waveform | 13.33 ±0.59 | 10.73 ±0.86 | 11.16 ±0.72 | 11.81 ±1.13 | 13.52 ±1.97 | 12.43 ±0.91 | 11.66 ±0.89 | 11.20 ±0.63 | 11.67 ±0.90 | 11.40 ±0.86 | 11.69 ±1.34 | 11.42 ±0.94 | 11.45 ±0.93 | 12.38 ±0.59 | 12.04 ±0.72 |
| german | 23.63 ±1.70 | 23.30 ±2.51 | 23.67 ±2.28 | 25.13 ±3.73 | 23.57 ±2.15 | 27.83 ±3.60 | 28.47 ±3.00 | 25.73 ±2.62 | 23.57 ±2.43 | 23.80 ±2.22 | 23.77 ±2.27 | 23.77 ±2.52 | 23.90 ±2.56 | 26.87 ±2.60 | 23.93 ±2.01 |
| image | 17.53 ±1.05 | 2.84 ±0.52 | 3.82 ±0.59 | 3.82 ±0.87 | 3.59 ±0.71 | 4.83 ±1.51 | 2.54 ±0.45 | 2.44 ±0.56 | 3.32 ±0.59 | 2.43 ±0.51 | 2.30 ±0.40 | 2.36 ±0.54 | 2.30 ±0.52 | 2.18 ±0.41 | 2.27 ±0.36 |
| ijcnn1 | 8.41 ±0.60 | 4.01 ±0.53 | 5.37 ±1.40 | 4.82 ±1.99 | 4.83 ±1.14 | 6.35 ±1.33 | 5.17 ±0.83 | 4.17 ±0.97 | 5.46 ±1.32 | 4.33 ±0.84 | 4.09 ±0.76 | 4.33 ±0.84 | 4.06 ±0.84 | 4.18 ±0.73 | 4.05 ±0.68 |
| a9a | 15.34 ±0.11 | 15.87 ±0.33 | 15.39 ±0.12 | 15.97 ±0.23 | 15.56 ±0.23 | 15.82 ±0.23 | 16.16 ±0.22 | 16.97 ±0.55 | 15.74 ±0.12 | 15.64 ±0.10 | 15.70 ±0.09 | 15.74 ±0.12 | 15.74 ±0.12 | 19.90 ±0.30 | 17.13 ±0.19 |
| Mean of SVM normalized errors | 2.237 | 1.000 | 1.110 | 1.234 | 1.227 | 1.260 | 1.087 | 1.031 | 1.219 | 1.009 | 0.998 | 1.006 | 0.996 | 1.073 | 1.029 |
| Mean of SVM normalized $K$ | 0.006 | 1.000 | 0.113 | 0.069 | 0.046 | 0.073 | 0.635 | 8.050 | 0.042 | 0.060 | 0.160 | 0.057 | 0.064 | 0.128 | 0.088 |

Table 3: The inferred depth of PBDN that increases its depth until a model selection criterion starts to rise.

| Dataset | banana | breast cancer | titanic | waveform | german | image | ijcnn1 | a9a |
|---|---|---|---|---|---|---|---|---|
| AIC-Gibbs | 2.30 ± 0.48 | 1.00 ± 0.00 | 1.00 ± 0.00 | 1.90 ± 0.74 | 1.30 ± 0.67 | 2.40 ± 0.52 | 2.00 ± 0.00 | 1.00 ± 0.00 |
| AIC$_{\epsilon=0.01}$-Gibbs | 2.30 ± 0.48 | 1.00 ± 0.00 | 1.00 ± 0.00 | 2.00 ± 0.67 | 1.60 ± 0.84 | 2.60 ± 0.52 | 3.40 ± 0.55 | 1.00 ± 0.00 |
| AIC-SGD | 3.20 ± 0.78 | 1.90 ± 0.99 | 1.00 ± 0.00 | 2.40 ± 0.52 | 2.80 ± 0.63 | 2.90 ± 0.74 | 3.20 ± 0.45 | 3.20 ± 0.45 |
| AIC$_{\epsilon=0.01}$-SGD | 2.80 ± 0.63 | 1.00 ± 0.00 | 1.00 ± 0.00 | 1.50 ± 0.53 | 1.00 ± 0.00 | 2.00 ± 0.00 | 3.00 ± 0.00 | 1.00 ± 0.00 |

We record the out-of-sample-prediction errors and computational complexity of various algorithms over these eight benchmark datasets in Tab. 2 and Tab. 5 of the Appendix, respectively, and summarize in Tab. 2 the means of SVM normalized errors and numbers of support hyperplanes/vectors. Overall, PBDN using AIC$_\epsilon$ in (10) with $\epsilon = 0.01$ to determine the depth, referred to as PBDN-AIC$_{\epsilon=0.01}$, has the highest out-of-sample prediction accuracy, followed by PBDN4, the RBF kernel SVM, PBDN using AIC in (9) to determine the depth, referred to as PBDN-AIC, PBDN2, PBDN-AIC$_{\epsilon=0.01}$ solved with SGD, DNN (128-64), PBDN-AIC solved with SGD, and DNN (32-16).

Overall, logistic regression does not perform well, which is not surprising as it is a linear classifier that uses a single hyperplane to partition the covariate space into two halves to separate one class from the other. As shown in Tab. 2 of the Appendix, for breast cancer, titanic, german, and a9a, all classifiers have comparable classification errors, suggesting minor or no advantages of using a nonlinear classifier on them. By contrast, for banana, waveform, image, and ijcnn1, all nonlinear classifiers clearly outperform logistic regression. Note PBDN1, which clearly reduces the classification errors of logistic regression, performs similarly to both AMM and CPM. These results are not surprising as CPM, closely related to AMM, uses a convex polytope, defined as the intersection of multiple hyperplanes, to enclose one class, whereas the classification decision boundaries of PBDN1 can be bounded within a convex polytope that encloses negative examples. Note that the number of hyperplanes are automatically inferred from the data by PBDN1, thanks to the inherent shrinkage mechanism of the gamma process, whereas the ones of AMM and CPM are both selected via cross validations. While PBDN1 can partially remedy their sensitivity to how the data are labeled by combining the results obtained under two opposite labeling settings, the decision boundaries of the two iSHMs and those of both AMM and CPM are still restricted to a confined space related to a single convex polytope, which may be used to explain why on banana, image, and ijcnn1, they all clearly underperform a PBDN with more than one hidden layer.

As shown in Tab. 2, DNN (8-4) clearly underperforms DNN (32-16) in terms of classification accuracy on both image and ijcnn1, indicating that having 8 and 4 hidden units for the first and second hidden layers, respectively, is far from enough for DNN to provide a sufficiently high nonlinear modeling capacity for these two datasets. Note that the equivalent number of hyperplanes for DNN $(K_1, K_2)$, a two-hidden-layer DNN with $K_1$ and $K_2$ hidden units in the first and second hidden layers, respectively, is computed as $[(V + 1)K_1 + K_1K_2]/(V + 1)$. Thus the computational complexity quickly increases as the network size increases. For example, DNN (8-4) is comparable to PBDN1 and PBDN-AIC in terms of out-of-sample-prediction computational complexity, as shown in Tabs.

2 and 5, but it clearly underperforms all of them in terms of classification accuracy, as shown in Tab. 2. While DNN (128-64) performs well in terms of classification accuracy, as shown in Tab. 2, its out-of-sample-prediction computational complexity becomes clearly higher than the other algorithms with comparable or better accuracy, such as RVM and PBDN, as shown in Tab. 5. In practice, however, the search space for a DNN with two or more hidden layers is enormous, making it difficult to determine a network that is neither too large nor too small to achieve a good compromise between fitting the data well and having low complexity for both training and out-of-sample prediction. E.g., while DNN (128-64) could further improve the performance of DNN (32-16) on these two datasets, it uses a much larger network and clearly higher computational complexity for out-of-sample prediction.

We show the inferred number of active support hyperplanes by PBDN in a single random trial in Figs. 3-6. For PBDN, the computation in both training and out-of-sample prediction also increases in $T$, the network depth. It is clear from Tab. 2 that increasing $T$ from 1 to 2 generally leads to the most significant improvement if there is a clear advantage of increasing $T$, and once $T$ is sufficiently large, further increasing $T$ leads to small performance fluctuations but does not appear to lead to clear overfitting. As shown in Tab. 3, the use of the AIC based greedy model selection criterion eliminates the need to tuning the depth $T$, allowing it to be inferred from the data. Note we have tried stacking CPMs as how we stack iSHMs, but found that the accuracy often quickly deteriorates rather than improving. E.g., for CPMs with (2, 3, or 4) layers, the error rates become (0.131, 0.177, 0.223) on waveform, and (0.046, 0.080, 0.216) on image. The reason could be that CPM infers redundant unweighted hyperplanes that lead to strong multicollinearity for the covariates of deep layers.

Note on each given data, we have tried training a DNN with the same network architecture inferred by a PBDN. While a DNN jointly trains all its hidden layers, it provides no performance gain over the corresponding PBDN. More specifically, the DNNs using the network architectures inferred by PBDNs with AIC-Gibbs, AIC$_{\epsilon=0.01}$-Gibbs, AIC-SGD, and AIC$_{\epsilon=0.01}$-SGD, have the mean of SVM normalized errors as 1.047, 1.011, 1.076, and 1.144, respectively. These observations suggest the efficacy of the greedy-layer-wise training strategy of the PBDN, which requires no cross-validation.

For out-of-sample prediction, the computation of a classification algorithm generally increases linearly in the number of support hyperplanes/vectors. Using logistic regression with a single hyperplane for reference, we summarize the computation complexity in Tab. 2, which indicates that in comparison to SVM that consistently requires the most number of support vectors, PBDN often requires significantly less time for predicting the class label of a new data sample. For example, for out-of-sample prediction for the image dataset, as shown in Tab. 5, on average SVM uses about 212 support vectors, whereas on average PBDNs with one to five hidden layers use about 13, 16, 29, 50, and 64 hyperplanes, respectively, and PBDN-AIC uses about 22 hyperplanes, showing that in comparison to kernel SVM, PBDN could be much more computationally efficient in making out-of-sample prediction.

## 5 Conclusions

The infinite support hyperplane machine (iSHM), which interacts countably infinite non-negative weighted hyperplanes via a noisy-OR mechanism, is employed as the building unit to greedily construct a capacity-regularized parsimonious Bayesian deep network (PBDN). iSHM has an inductive bias in fitting the positively labeled data, and employs the gamma process to infer a parsimonious set of active hyperplanes to enclose negatively labeled data within a convex-polytope bounded space. Due to the inductive bias and label asymmetry, iSHMs are trained in pairs to ensure a sufficient coverage of the covariate space occupied by the data from both classes. The sequentially trained iSHM pairs can be stacked into PBDN, a feedforward deep network that gradually enhances its modeling capacity as the network depth increases, achieving high accuracy while having low computational complexity for out-of-sample prediction. PBDN can be trained using either Gibbs sampling that is suitable for quantifying posterior uncertainty, or SGD based MAP inference that is scalable to big data. One may potentially construct PBDNs for regression analysis of count, categorical, and continuous response variables by following the same three-step strategy: constructing a nonparametric Bayesian model that infers the number of components for the task of interest, greedily adding layers one at a time, and using a forward model selection criterion to decide how deep is deep enough. For the first step, the recently proposed Lomax distribution based racing framework [34] could be a promising candidate for both categorical and non-negative response variables, and Dirichlet process mixtures of generalized linear models [35] could be promising candidates for continuous response variables and many other types of variables via appropriate link functions.

**Acknowledgments**

M. Zhou acknowledges the support of Award IIS-1812699 from the U.S. National Science Foundation, the support of NVIDIA Corporation with the donation of the Titan Xp GPU used for this research, and the computational support of Texas Advanced Computing Center.

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
