[Supplementary Material]

# Parsimonious Bayesian deep networks: supplementary material

Mingyuan Zhou

---

**Algorithm 1:** Gibbs sampling (*MAP via SGD*) for iSHM.

**Inputs:** $y_i$: the observed labels, $\boldsymbol{x}_i$: covariate vectors, $K_{\max}$: the upper-bound of the number of hyperplanes, $I_{Prune}$: the set of iterations at which the hyperplanes with $\sum_i b_{ik} = 0$ are pruned.

**Outputs:** the maximum likelihood sample (*MAP solution*) that includes $K$ active support hyperplanes $\boldsymbol{\beta}_k$ and their weights $r_k$, where hyperplane $k$ is defined as active if $\sum_i b_{ik} > 0$.

---

1: Initialize the model parameters with $\boldsymbol{\beta}_k = 0$, $K = K_{\max}$, and $r_k = 1/K_{\max}$.
2: **for** $iter = 1 : maxIter$ **do**
3:   **if** Gibbs sampling **then**
4:     Sample $m_i$; Sample $\{m_{ik}\}_k$;
5:   **else**
6:     Retrive a mini batch
7:     **if** $iter \in I_{Prune}$ **then**
8:       Sample $b_{ik} \sim \text{Bernoulli}(p_{ik})$, $p_{ik} = 1 - e^{-r_k \ln(1+e^{\boldsymbol{\beta}'_k \boldsymbol{x}_i})}$
9:       **if** $y_i = 1$ and $\sum_k b_{ik} = 0$ **then**
10:        $(b_{i1}, \ldots, b_{iK}) \sim \text{Multinomial}\left(1, \frac{p_{i1}}{\sum_{k=1}^K p_{ik}}, \ldots, \frac{p_{iK}}{\sum_{k=1}^K p_{ik}}\right)$
11:      **end if**
12:    **end if**
13:  **end if**
14:  **for** $k = 1, \ldots, K$ **do**
15:    **if** Gibbs sampling **then**
16:      Sample $l_{ik}, \omega_{ik}, \boldsymbol{\beta}_k, b_{\beta k}, r_k$, and $\theta_{ik}$;
17:    **else**
18:      SGD update of $\boldsymbol{\beta}_k$ and $\ln r_k$
19:    **end if**
20:    **if** $iter \in I_{Prune}$ and $\sum_i b_{ik} = 0$ **then**
21:      Prune Expert $k$ and reduce $K$ by one
22:    **end if**
23:  **end for**
24: **end for**

---

**Algorithm 2:** Greedy layer-wise training for PBDN.

---

1: Denote $\boldsymbol{x}_i^{(1)} = \boldsymbol{x}_i$ and $\text{AIC}(T) = \infty$.
2: **for** Layer $t = 1 : \infty$ **do**
3:   Train an iSHM to predict $y_i$ given $\boldsymbol{x}_i^{(t)}$;
4:   Train an iSHM to predict $y_i^* = 1 - y_i$ given $\boldsymbol{x}_i^{(t)}$;
5:   Compute $P(y_i \,|\, \boldsymbol{x}_i^{(t)})$, $P(y_i^* \,|\, \boldsymbol{x}_i^{(t)})$, and $\text{AIC}(t)$;
6:   **if** $\text{AIC}(t) < \text{AIC}(t-1)$ **then**
7:     Combine two iSHMs to produce $\boldsymbol{x}_i^{(t+1)}$;
8:   **else**
9:     Use the first $(t-1)$ iSHM pairs to compute the conditional class probability $P(y_i \,|\, \boldsymbol{x}_i)$;
10:  **end if**
11: **end for**

---

## A  Proofs

*Proof of Theorem 1.* Since $\sum_{k' \neq k} r_{k'} \ln(1 + e^{\boldsymbol{x}'_i \boldsymbol{\beta}_{k'}}) \geq 0$ a.s., if (6) is true, then $r_k \ln(1 + e^{\boldsymbol{x}'_i \boldsymbol{\beta}_k}) \leq -\ln(1 - p_0)$ a.s. for all $k \in \{1, 2, \ldots\}$. Thus if (6) is true, then (7) is true a.s., which means the set of solutions to (6) is included in the set of solutions to (7). □

Figure 2: Graphical representation of (a) the generative model for iSHM, (b) the asymmetric conditional class probability function, and (c) a $T$-hidden-layer PBDN that stacks a pair of iSHMs after another to construct a feedforward deep network.

## B  Related multi-hyperplane models

Generalizing the construction of multiclass support vector machines in Crammer and Singer [36], the idea of combining multiple hyperplanes to define complex classification decision boundaries has been discussed before [26–30]. In particular, the convex polytope machine (CPM) [30] exploits the idea of learning a convex polytope to separate one class from the other. From this point of view, the proposed iSHM is related to the CPM as its decision boundary can be explicitly bounded by a convex polytope that encloses the data labeled as zeros, as described in Theorem 1 and illustrated in Fig. 1. Distinct from the CPM that uses a convex polytope as its decision boundary, and provides no probability estimates for class labels and no principled ways to set its number of equally-weighted hyperplanes, iSHM makes its decision boundary smoother than the corresponding bounding convex polytope, as shown in Figs. 1 (c) and (f), by using more complex interactions between hyperplanes than simple intersection. iSHM also provides probability estimates for its labels, and supports countably infinite differently-weighted hyperplanes with the gamma process. In addition, to solve its non-convex objective function, the CPM relies on heuristics to force the learning of each hyperplane as a convex optimization problem, whereas iSHM can use Bayesian inference, in which each data point assigns a binary indicator to each hyperplane. Moreover, iSHM pair is used as the building unit to construct PBDN to quickly boost the modeling power.

## C  Gibbs sampling update equations

To begin with, we sample the latent count $m_i$ and then partitions it into $m_{ik}$ for different hyperplanes, where the value of $m_i$ is related to how likely $y_i = 1$ in the posterior, and the ratio $m_{ik}/m_i$ is related to how much does expert $k$ contribute to the overall cause of $y_i = 1$. Below we first describe the Gibbs sampling update equations for $m_i$ and $m_{ik}$.

***Sample*** $m_i$. Denote $\theta_{i\cdot} = \sum_{k=1}^{K} \theta_{ik}$. Since $m_i = 0$ a.s. given $y_i = 0$ and $m_i \geq 1$ given $y_i = 1$, and in the prior $m_i \sim \text{Pois}(\theta_{i\cdot})$, following the inference in Zhou [18], we can sample $m_i$ as

$$(m_i \,|\, -) \sim y_i \text{Pois}_+ (\theta_i.) , \tag{11}$$

where $m \sim \text{Pois}_+(\theta)$ denotes a draw from the zero-truncated Poisson distribution.

***Sample*** $m_{ik}$. Once the latent counts $m_{ik}$ are known, it becomes clear on how much expert $k$ contributes to the cause of $y_i = 1$. Since letting $m_i = \sum_{k=1}^{K} m_{ik}$, $m_{ik} \sim \text{Pois}(\theta_{ik})$ is equivalent in distribution to letting $(m_{i1}, \ldots, m_{iK}) \,|\, m_i \sim \text{Mult}(m_i, \theta_{i1}/\theta_i., \ldots, \theta_{iK}/\theta_i.)$, $m_i \sim \text{Pois}(\theta_i.)$, similar to Dunson and Herring [37] and Zhou et al. [38], we sample $m_{ik}$ as

$$(m_{i1}, \ldots, m_{iK} \,|\, -) \sim \text{Mult}(m_i, \theta_{i1}/\theta_i., \ldots, \theta_{iK}/\theta_i.) . \tag{12}$$

The key remaining problem is to infer $\boldsymbol{\beta}_k$. Note that marginalizing out $\theta_{ik}$ from (5) leads to

$$m_{ik} \sim \text{NB}\big[r_k, 1/(1 + e^{-\boldsymbol{x}_i'\boldsymbol{\beta}_k})\big], \tag{13}$$

where $m \sim \text{NB}(r, p)$ represents a negative binomial (NB) distribution with shape $r$ and probability $p$. We thus exploit the augmentation techniques developed for the NB distribution in Zhou et al. [39] to sample $r_k$, and these developed for logistic regression in Polson et al. [40] and further generalized to NB regression in Zhou et al. [41] and Polson et al. [42] to sample $\boldsymbol{\beta}_k$. We outline Gibbs sampling in Algorithm 1, where to save computation, we set $K_{\max}$ as the upper-bound of the number of experts and prune the experts assigned with zero counts during MCMC iterations. Note that except for the sampling of $\{m_{ik}\}_k$, the sampling of all the other parameters of different experts are embarrassingly parallel.

Gibbs sampling via data augmentation and marginalization proceeds as follows.

***Sample*** $\boldsymbol{\beta}_k$. Using data augmentation for NB regression, as in Zhou et al. [41] and [42], we denote $\omega_{ik}$ as a random variable drawn from the Polya-Gamma (PG) distribution [40] as $\omega_{ik} \sim \text{PG}(m_{ik} + \theta_{ik}, 0)$, under which we have $\mathbb{E}_{\omega_{ik}}\big[\exp(-\omega_{ik}(\psi_{ik})^2/2)\big] = \cosh^{-(m_{ik}+r_k)}(\psi_{ik}/2)$. Since $m_{ik} \sim \text{NB}\big[r_k, 1/(1 + e^{-\boldsymbol{x}_i'\boldsymbol{\beta}_k})\big]$, the likelihood of $\psi_{ik} := \boldsymbol{x}_i\boldsymbol{\beta}_k$ can be expressed as

$$
\begin{aligned}
\mathcal{L}(\psi_{ik}) &\propto \frac{(e^{\psi_{ik}})^{m_{ik}}}{(1 + e^{\psi_{ik}})^{m_{ik}+\theta_{ik}}} \\
&= \frac{2^{-(m_{ik}+\theta_{ik})} \exp(\frac{m_{ik}-\theta_{ik}}{2}\psi_{ik})}{\cosh^{m_{ik}+\theta_{ik}}(\psi_{ik}/2)} \\
&\propto \exp\left(\frac{m_{ik}-\theta_{ik}}{2}\psi_i\right) \mathbb{E}_{\omega_{ik}}\big[\exp[-\omega_{ik}(\psi_{ik})^2/2]\big] .
\end{aligned}
$$

Combining the likelihood $\mathcal{L}(\psi_{ik}, \omega_{ik}) \propto \exp\left(\frac{m_{ik}-\theta_{ik}}{2}\psi_i\right) \exp[-\omega_{ik}(\psi_{ik})^2/2]$ and the prior, we sample auxiliary Polya-Gamma random variables $\omega_{ik}$ as

$$(\omega_{ik} \,|\, -) \sim \text{PG}(m_{ik} + r_k, \, \boldsymbol{x}_i'\boldsymbol{\beta}_k) , \tag{14}$$

conditioning on which we sample $\boldsymbol{\beta}_k$ as

$$
\begin{aligned}
(\boldsymbol{\beta}_k \,|\, -) &\sim \mathcal{N}(\boldsymbol{\mu}_k, \boldsymbol{\Sigma}_k), \\
\boldsymbol{\Sigma}_k &= \left(\text{diag}(\alpha_{1k}, \ldots, \alpha_{Vk}) + \sum_i \omega_{ik}\boldsymbol{x}_i\boldsymbol{x}_i'\right)^{-1}, \\
\boldsymbol{\mu}_k &= \boldsymbol{\Sigma}_k \left[\sum_i \left(\frac{m_{ik}-r_k}{2}\right) \boldsymbol{x}_i\right].
\end{aligned} \tag{15}
$$

Note to sample from the Polya-Gamma distribution, we use a fast and accurate approximate sampler of Zhou [43] that matches the first two moments of the true distribution; we set the truncation level of that sampler as five.

***Sample*** $\theta_{ik}$. Using the gamma-Poisson conjugacy, we sample $\theta_{ik}$ as

$$(\theta_{ik} \,|\, -) \sim \text{Gamma}\left(r_k + m_{ik}, \frac{e^{\boldsymbol{x}_i'\boldsymbol{\beta}_k}}{1 + e^{\boldsymbol{x}_i'\boldsymbol{\beta}_k}}\right). \tag{16}$$

***Sample*** $\alpha_{vk}$. We sample $\alpha_{vk}$ as

$$(\alpha_{vk} \mid -) \sim \text{Gamma}\left(a_\beta + \frac{1}{2}, \frac{1}{b_{\beta k} + \frac{1}{2}\beta_{vk}^2}\right). \tag{17}$$

***Sample $b_{\beta k}$.*** We sample $b_{\beta k}$ as

$$(b_{\beta k} \mid -) \sim \text{Gamma}\left(e_0 + a_\beta(V+1), \frac{1}{f_0 + \sum_v \alpha_{vk}}\right). \tag{18}$$

***Sample $c_0$.*** We sample $c_0$ as

$$(c_0 \mid -) \sim \text{Gamma}\left(e_0 + \gamma_0, \frac{1}{f_0 + \sum_k r_k}\right). \tag{19}$$

***Sample $l_{ik}$.*** We sample $l_{ik}$ using the Chinese restaurant table (CRT) distribution [39] as

$$(l_{ik} \mid -) \sim \text{CRT}(m_{ik}, r_k). \tag{20}$$

***Sample $\gamma_0$ and $r_k$.*** Let us denote

$$\tilde{p}_k := \sum_i \ln(1 + e^{\boldsymbol{x}_i'\boldsymbol{\beta}_k}) \Big/ \Big[c_0 + \sum_i \ln(1 + e^{\boldsymbol{x}_i'\boldsymbol{\beta}_k})\Big].$$

Given $l_{\cdot k} = \sum_i l_{ik}$, we first sample

$$(\tilde{l}_k \mid -) \sim \text{CRT}(l_{\cdot k}, \gamma_0/K). \tag{21}$$

With these latent counts, we then sample $\gamma_0$ and $r_k$ as

$$(\gamma_0 \mid -) \sim \text{Gamma}\left(a_0 + \tilde{l}_\cdot, \frac{1}{b_0 - \frac{1}{K}\sum_k \ln(1 - \tilde{p}_k)}\right),$$
$$(r_k \mid -) \sim \text{Gamma}\left(\frac{\gamma_0}{K} + l_{\cdot k}, \frac{1}{c_0 + \ln(1 + e^{\boldsymbol{x}_i'\boldsymbol{\beta}_k})}\right). \tag{22}$$

## D   Experimental settings and additional results

Following Tipping [31], we consider the following datasets: banana, breast cancer, titanic, waveform, german, and image. For each of these six datasets, we consider the first ten predefined random training/testing partitions, and report both the sample mean and standard deviation of the testing classification errors. Since these datasets, originally provided by Rätsch et al. [44], were no longer available on the authors' websites, we use the version provided by Diethe [45]. We also consider two additional benchmark datasets: ijcnn1 and a9a [27, 30, 46]. Instead of using a fixed training/testing partition that comes with ijcnn1 and a9a, for a more rigorous comparison, we use the $(i, i+10, \ldots)$th observations as training and the remaining ones as testing, and run five independent random trials with $i \in \{1, 2, 3, 4, 5\}$.

We use the $L_2$ regularized logistic regression provided by the LIBLINEAR package [47] to train a linear classifier, where a bias term is included and the regularization parameter $C$ is five-fold cross-validated on the training set from $(2^{-10}, 2^{-9}, \ldots, 2^{15})$.

For kernel SVM, a Gaussian RBF kernel is used and three-fold cross validation is used to tune both the regularization parameter $C$ and kernel width on the training set. We use the LIBSVM package [48], where we three-fold cross-validate both the regularization parameter $C$ and kernel-width parameter $\gamma$ on the training set from $(2^{-5}, 2^{-4}, \ldots, 2^5)$, and choose the default settings for all the other parameters.

For RVM, instead of directly quoting the results from Tipping [31], which only reported the mean but not standard deviation of the classification errors for each of the first six datasets in Tab. 4, we use the matlab code[1] provided by the author, using a Gaussian RBF kernel whose kernel width is three-fold cross-validated on the training set from $(2^{-5}, 2^{-4.5}, \ldots, 2^5)$ for both ijcnn1 and a9a and from $(2^{-10}, 2^{-9.5}, \ldots, 2^{10})$ for all the others.

We consider adaptive multi-hyperplane machine (AMM) [27], as implemented in the BudgetSVM[2] (Version 1.1) software package [49]. We consider the batch version of the algorithm. Important parameters of the AMM include both the regularization parameter $\lambda$ and training epochs $E$. As also observed in Kantchelian et al. [30], we do not observe the testing errors of AMM to strictly decrease as $E$ increases. Thus, in addition to cross validating the regularization parameter $\lambda$ on the training set from $\{10^{-7}, 10^{-6}, \ldots, 10^{-2}\}$, as done in Wang et al. [27], for each $\lambda$, we try $E \in \{5, 10, 20, 50, 100\}$ sequentially until the cross-validation error begins to decrease, *i.e.*, under the same $\lambda$, we choose $E = 20$ if the cross-validation error of $E = 50$ is greater than that of $E = 20$. We use the default settings for all the other parameters.

Table 4: Binary classification datasets used in experiments, where $V$ is the feature dimension.

| Dataset | banana | breast cancer | titanic | waveform | german | image | ijcnn1 | a9a |
|---|---|---|---|---|---|---|---|---|
| Train size | 400 | 200 | 150 | 400 | 700 | 1300 | 14,169 | 4,884 |
| Test size | 4900 | 77 | 2051 | 4600 | 300 | 1010 | 127,522 | 43,958 |
| $V$ | 2 | 9 | 3 | 21 | 20 | 18 | 22 | 123 |

We consider the convex polytope machine (CPM) [30], using the python code[3] provided by the authors. Important parameters of the CPM include the entropy parameter $h$, regularization factor $C$, and number of hyperplanes $K$ for each side of the CPM ($2K$ hyperplanes in total). Similar to the setting of Kantchelian et al. [30], we first fix $h = 0$ and select the best regularization factor $C$ from $\{10^{-4}, 10^{-3}, \ldots, 10^0\}$ using three-fold cross validation on the training set. For each $C$, we try $K \in \{1, 3, 5, 10, 20, 40, 60, 80, 100\}$ sequentially until the cross-validation error begins to decrease. With both $\lambda$ and $K$ selected, we then select $h$ from $\{0, \ln(K/10), \ln(2K/10), \ldots, \ln(9K/10)\}$. For each trial, we consider 10 million iterations in cross-validation and 32 million iterations in training with the cross-validated parameters. Note different from Kantchelian et al. [30], which suggests that the error rate decreases as $K$ increases, we cross-validate $K$ as we have found that the testing errors of the CPM may increase once it increases over certain limits.

Table 5: Analogous table to Tab. 2 that shows the comparison of the (equivalent) number of support vectors/hyperplanes between various algorithms.

| | LR | SVM | RVM | AMM | CPM | DNN (8-4) | DNN (32-16) | DNN (128-64) | PBDN1 | PBDN2 | PBDN4 | AIC Gibbs | AIC$_\epsilon$ Gibbs | AIC SGD | AIC$_\epsilon$ SGD |
|---|---|---|---|---|---|---|---|---|---|---|---|---|---|---|---|
| banana | 1.0 ±0.0 | 129.2 ±32.8 | 22.3 ±26.0 | 9.5 ±2.8 | 14.2 ±7.9 | 18.7 ±0.0 | 202.7 ±0.0 | 2858.7 ±0.0 | 7.6 ±2.6 | 9.4 ±1.1 | 17.2 ±3.6 | 10.0 ±1.1 | 10.0 ±1.1 | 57.5 ±19.1 | 45.9 ±15.2 |
| breast cancer | 1.0 ±0.0 | 115.1 ±11.2 | 24.8 ±28.3 | 13.4 ±0.8 | 5.2 ±3.7 | 11.2 ±0.0 | 83.2 ±0.0 | 947.2 ±0.0 | 7.1 ±2.8 | 9.1 ±1.5 | 24.4 ±6.0 | 7.1 ±2.8 | 7.1 ±2.8 | 18.2 ±10.3 | 6.5 ±1.0 |
| titanic | 1.0 ±0.0 | 83.4 ±13.3 | 5.1 ±3.0 | 14.9 ±3.1 | 4.8 ±3.3 | 16.0 ±0.0 | 160.0 ±0.0 | 2176.0 ±0.0 | 4.2 ±0.4 | 7.0 ±2.5 | 12.8 ±1.3 | 4.2 ±0.4 | 4.2 ±0.4 | 4.2 ±0.4 | 4.2 ±0.4 |
| waveform | 1.0 ±0.0 | 147.0 ±38.5 | 21.1 ±11.0 | 9.5 ±1.2 | 4.4 ±2.8 | 9.5 ±0.0 | 55.3 ±0.0 | 500.4 ±0.0 | 5.5 ±1.7 | 10.6 ±2.1 | 35.1 ±6.3 | 12.5 ±7.6 | 12.3 ±8.2 | 10.4 ±1.8 | 7.2 ±2.6 |
| german | 1.0 ±0.0 | 423.6 ±55.0 | 11.0 ±3.2 | 18.8 ±1.8 | 2.8 ±1.7 | 9.5 ±0.0 | 56.4 ±0.0 | 518.1 ±0.0 | 8.2 ±2.3 | 14.1 ±4.5 | 40.1 ±7.9 | 10.2 ±6.0 | 12.1 ±7.6 | 46.4 ±10.5 | 15.6 ±1.3 |
| image | 1.0 ±0.0 | 211.6 ±47.5 | 35.8 ±9.2 | 10.5 ±1.1 | 14.6 ±7.5 | 9.7 ±0.0 | 58.9 ±0.0 | 559.2 ±0.0 | 12.9 ±1.2 | 16.4 ±1.5 | 50.4 ±3.1 | 21.7 ±5.9 | 24.1 ±6.6 | 22.4 ±2.9 | 19.4 ±2.2 |
| ijcnn1 | 1.0 ±0.0 | 835.2 ±139.5 | 83.4 ±45.2 | 8.2 ±0.8 | 38.0 ±8.4 | 9.4 ±0.0 | 54.3 ±0.0 | 484.2 ±0.0 | 28.4 ±0.9 | 33.9 ±1.4 | 89.5 ±2.6 | 33.9 ±1.4 | 67.3 ±19.5 | 45.0 ±13.2 | 41.6 ±7.3 |
| a9a | 1.0 ±0.0 | 1884.2 ±79.7 | 26.2 ±4.8 | 28.0 ±3.9 | 2.8 ±1.8 | 8.3 ±0.0 | 36.1 ±0.0 | 194.1 ±0.0 | 24.4 ±3.9 | 39.1 ±6.1 | 198.7 ±23.9 | 24.4 ±3.9 | 24.4 ±3.9 | 48.1 ±1.9 | 26.0 ±1.4 |
| Mean of SVM normalized $K$ | 0.006 | 1.000 | 0.113 | 0.069 | 0.046 | 0.073 | 0.635 | 8.050 | 0.042 | 0.060 | 0.160 | 0.057 | 0.064 | 0.128 | 0.088 |

[2] http://www.dabi.temple.edu/budgetedsvm/
[3] https://github.com/alkant/cpm

Figure 3: The inferred weights of the active experts (support hyperplanes) of iSHM pair of the single-hidden-layer deep softplus network (PBDN-1), ordered from left to right according to their weights, on six benchmark datasets, based on the maximum likelihood sample of a single random trial.

Figure 4: Analogous to Fig. 3 for the most recently added iSHM pair of the two-hidden-layer deep softplus network (PBDN-2).

Figure 5: Analogous to Fig. 3 for the most recently added iSHM pair of the three-hidden-layer deep softplus network (PBDN-3).

Figure 6: Analogous to Fig. 3 for the most recently added iSHM pair of the four-hidden-layer deep softplus network (PBDN-4).

## Footnotes

[1]`http://www.miketipping.com/downloads/SB2_Release_200.zip`