[Reviews · NeurIPS 2018]

Reviewer 1



The authors propose an algorithm for constructing deep neural networks adapted to the complexity of a binary classification problem. As a Bayesian probabilistic model is used for each layer, its width can be learned from the examples using Gibbs sampling or maximum a posteriori inference. New layers are added greedily until the two classes are well separated. Results show that this method achieves similar error rates as state-of-the-art classification algorithms, but the complexity of the constructed network indicated by the number of hidden units is often significantly smaller. The paper is well written. It explains the proposed method for constructing the network clearly and discusses the results at length. The infinite support hyper-plane machine (iSHM), which is used as layer here, is an interesting and novel application of non-parametric Bayesian inference, which enables learning of the network structure directly without cross-validation. This property is a big advantage compared to conventional deep neural networks and consequently the parsimonious Bayesian deep network may find a wide use as an automatically adapting classification algorithm. I have read the feedback provided by the authors.

Reviewer 2



The paper introduces a new type of (deep) neural network for binary classification. Each layer is in principle infinitely wide but in practice finite number of units is used. The layers are trained sequentially by first training one layer, and then always the next layer after the previous one. The main claim is that the proposed model gives comparable results to the alternative approaches by utilizing fewer hyperplanes that results in faster out-of-sample predictions. The approach seems somewhat novel and the results support the claim to some extent. However, I found the paper difficult to read in many places (some specific comments below). Furthermore, the accuracy is essentially the same as for the other methods, the computational cost at the training time is not considered and other techniques used to reduce computations at prediction time are not discussed (see below). My overall feeling is that the paper has some potential, but the current exposition of the material requires too much effort from the reader, and hence the paper does not quite meet the standards of NIPS. Some specific comments/questions: - I felt the introduction of the iSHM is unnecessarily complicated. The model is actually far more simple than made to appear. It feels that Section 2 could have been both compressed and made clearer simultaneously, as now it contains many not-so-important mathematical details that simply confuse the reader rather than clarify things. The same applies also to Section 3 to some extent. - All the equations have too little space which makes reading the text and formulae somewhat uneasy. - I found Fig. 1 extremely hard to follow. Not only are the figures so small that they are barely readable, but the subfigure labeling is also confusing; the outer subfigure names are 1), 2), c) and d) (why not 1-4?), and these are also divided further into 'subsubfigures' named a)-f). It took far too much effort to try to understand the message of these figures. Furthermore, this figure is not even discussed in the main text: "We provide detailed explanations for Fig. 1 in Appendix D." Partly due to these reasons I never got a good intuition about how the model actually works and why it would be better than the traditional ones. - I would expect the proposed model to be outperformed by convolutional networks (CNN) in image classification. This is not to say that it wouldn't be useful in some other settings, but since the authors claim (in the abstract) that the method achieves "state-of-the-art classification accuracy", I'm simply curious whether the authors agree about this intuition? - The authors focus on the computational complexity for out-of-sample predictions. What is the computational cost at training time compared to the more traditional networks (e.g. those you used in the comparisons)? - If the purpose is to reduce the computations at prediction time, there are also other methods specifically designed for this, namely those that attempt to replace the complex architecture by a simpler one, such that the simple model utilizes information from the complex one [e.g. 1, 2]. To my knowledge, these have been quite successful and would be interesting to know how your method compares to these approaches. - Is it straightforward to extend the proposed model to multiclass problems or regression? Typos / other minor points: - 'nueral' page 2, line 63 - abbreviation iSHM introduced twice (page 2, first on line 57 and second time on line 81) - 'refereed' -> 'referred'? page 7, line 252 References: [1] Cristian Bucilu, Rich Caruana, and Alexandru Niculescu-Mizil. Model compression. In Proceedings of the 12th ACM SIGKDD International Conference on Knowledge Discovery and Data Mining, pages 535–541. ACM, 2006. [2] Geoffrey Hinton, Oriol Vinyals, Jeff Dean. Distilling the Knowledge in a Neural Network. https://arxiv.org/abs/1503.02531 *** Update *** I thank the authors for submitting a rebuttal which addressed all my questions. After taking some more time to read the paper again I've changed my mind and now think that this could be worthwhile contribution to the conference. I have updated my overall score accordingly. I do, however, encourage the authors to take my initial comments - especially those regarding the readability - into account when preparing the final version of the paper.

Reviewer 3



The paper proposes a generic way to stack Bayesian nonparametric models for deep learning. The idea is to use a Gamma process to express necessary hyperplanes together with their weights. Thus one layer looks like a noisy-or model. To stack multiple layers, the author proposes a forward selection method. To add a layer, they pass the linear predictors for each hyperplane to the next layer and double the predictors by flipping the output. The result shows BNP method can autpmatically learn economic representation of neural networks. This paper is very well written and it does not seem formidable to look into the details of their model. It is non-trivial to make deep generative models work at a level shown in the paper. Still I think Gibbs sampling can be slower than automatic differentiation methods. I wonder how Gibbs sampling works when your network depth become large? A relative line of work on using mixture of experts are mixtures of generalized linear models (such as logistic regression), where the mixing distribution is modeled as a Dirichlet process. For example, there's a paper named "Dirichlet process mixtures of generalized linear models". How do you compare your noisy-or model with those methods, for one layer and the potential to stack multiple layers? I like this paper. It has a solid contribution to (nonparametric) Bayesian deep learning. And it provides a novel way to do model compression for deep learning via Bayesian methods.